# Coronavirus Disease 2019-Associated Thrombotic Microangiopathy: A Single-Center Experience

**DOI:** 10.3390/ijms252212475

**Published:** 2024-11-20

**Authors:** Marija Malgaj Vrečko, Andreja Aleš-Rigler, Špela Borštnar, Željka Večerić-Haler

**Affiliations:** 1Department of Nephrology, University Medical Center Ljubljana, 1000 Ljubljana, Sloveniazeljka.vecerichaler@kclj.si (Ž.V.-H.); 2Faculty of Medicine, University of Ljubljana, 1000 Ljubljana, Slovenia

**Keywords:** thrombotic microangiopathy, thrombotic thrombocytopenic purpura, atypical hemolytic uremic syndrome, COVID-19, acute kidney injury, pathophysiology

## Abstract

Coronavirus disease 2019 (COVID-19) can lead to various multisystem disorders, including thrombotic microangiopathy (TMA). We present here eight patients with COVID-19-associated TMA who were treated at our center. Our aim was to summarize the demographic and clinical characteristics of the patients and discuss the possible role of COVID-19. One patient presented with thrombotic thrombocytopenic purpura (TTP) and seven with atypical hemolytic–uremic syndrome (aHUS.) Most patients had no obvious symptoms of COVID-19, and TMA occurred after viremia. Two patients had concomitant non-COVID-19-related triggers for TMA: exposure to tacrolimus and everolimus; first presentation of antiphospholipid syndrome. The patient with TTP was treated with therapeutic plasma exchange (TPE), steroids and caplacizumab, resulting in complete hematologic recovery. Six patients with aHUS were treated with TPE with or without steroids, four of whom received a C5 complement inhibitor and one an intravenous immunoglobulin. One patient with aHUS was treated with a C5 complement inhibitor and a steroid. We observed one partial and one complete recovery of renal function, while five patients experienced renal failure. There were no deaths. We believe that COVID-19 may act as a trigger for TMA in patients who have either pre-existing endothelial injury or an underlying predisposition to complement activation, and may also trigger autoimmune diseases. As a consequence of the different underlying pathophysiologies, the treatment of COVID-19-associated TMA requires a specific approach based on the subtype of the syndrome and possible concomitant triggers.

## 1. Introduction

Coronavirus disease 2019 (COVID-19) can lead to various multisystemic disorders that also affect the kidneys [1]. Although acute kidney injury (AKI) is common in patients with COVID-19 [2], the exact mechanism is still unclear. The most commonly reported histopathological findings on renal biopsy from patients with AKI associated with COVID-19 are acute tubular injury and focal segmental glomerular sclerosis (FSGS) [3,4]. However, there is evidence that thrombotic microangiopathy (TMA) can also occur [5].

TMA is not a single disease entity, but a group of extremely diverse syndromes that can be hereditary or acquired and can occur suddenly or gradually in children or adults [6]. The primary types of TMA include thrombotic thrombocytopenic purpura (TTP), which results from an inherited or acquired deficiency of ADAMTS13, typical hemolitic uremic syndrome, which is caused by Shiga toxin-producing Escherichia coli (STEC-HUS), and all other forms, which are grouped under the common term atypical hemolitic uremic syndrome (aHUS) [7].

Despite their diversity, all TMA syndromes have important pathological and clinical features in common. The pathologic hallmark of TMA is widespread microvascular injury manifested by platelet and fibrin thrombi in capillaries and arterioles with characteristic abnormalities of the endothelium and vessel wall. The classic clinical triad of TMA includes microangiopathic hemolytic anemia (MAHA), thrombocytopenia and organ damage (AKI, neurological abnormalities) [6]. The clinical picture of AKI reflects the consequences of ischemia in the kidney [8].

The emergence of COVID-19, which is caused by severe acute respiratory syndrome coronavirus 2 (SARS-CoV-2), has drawn considerable attention to TMA as a complication of the disease [5,9]. In 2022, we published a literature review on COVID-19-associated TMA, which revealed 46 cases of COVID-19-associated TMA in adult patients, of which 18 presented with TTP and 28 with aHUS [10]. The exact role of COVID-19 in the development of this TMA is not yet fully understood. Consequently, the heterogeneity of the treatment approach is due to the different and poorly understood underlying pathophysiologies.

Considering the diversity and complexity of COVID-19-associated TMA, we analyzed the data of patients from our center who were treated for COVID-19-associated TMA until August 2024. We present here a series of eight patients with COVID-19-associated TMA with their demographic and clinical characteristics, treatments and outcomes.

## 2. Results

### 2.1. Demographic and Clinical Characteristics of Patients with COVID-19-Associated TMA

Based on the Goodship et al. [7] classification used for this analysis, one patient presented with TTP and seven presented with aHUS. The main demographic and clinical characteristics of the patients are summarized in Table 1, and detailed information on the laboratory data and clinical course of patients is provided in the Appendix A.

#### 2.1.1. Patient Presenting with TTP

The only patient who presented with TTP was a 35-year-old woman with a known history of TTP. ADAMTS13 activity was only 1.7%, and we were also able to detect antibodies against ADAMTS13. The clinical presentation of COVID-19 was asymptomatic. At the time of TTP presentation, she had a positive polymerase chain reaction (PCR)-based smear for COVID-19. She complained of headaches, while no other neurological abnormalities were detected. Her lowest platelet count was 37 × 10^9^/L, and her renal function was intact from the onset of clinical presentation. She was treated with therapeutic plasma exchange (TPE) with fresh frozen plasma (FFP) in combination with steroids and caplacizumab. The treatment resulted in complete hematologic recovery.

#### 2.1.2. Patients Presenting with aHUS

Seven patients presented with aHUS. They had severe AKI and hemolytic anemia. Renal biopsy was performed in six patients, all of whom had pathohistological signs of TMA, two of them with additional features of concomitant collapsing FSGS. Patients had a median platelet count of 58 × 10^9^/L (range 25–127).

Most patients with aHUS (n = 6) had no obvious symptoms of COVID-19. Only one patient (*patient 3*) had a mild clinical presentation of COVID-19 with fatigue, fever and nausea, but without respiratory symptoms. All patients with aHUS developed signs of TMA after clearance of the virus (i.e., they had a negative PCR-based nasopharyngeal swab but positive serology indicating recent COVID-19 infection).

Three patients with aHUS had previously known comorbidities. *Patient 6* was a kidney transplant recipient (KTR) with end-stage renal disease (ESRD) due to IgA glomerulonephritis and had diabetes mellitus and arterial hypertension. He was receiving treatment with tacrolimus and everolimus. *Patient 4* had untreated arterial hypertension and congenital pulmonary valve stenosis, and *patient 7* had congenital hearing loss. In one patient (*patient 2*), we found positive lupus anticoagulants and anti-cardiolipin antibodies, and he was later diagnosed with antiphospholipid syndrome (aPS). On renal biopsy of two patients (*patients 7* and *8*), there were signs of previously unknown IgA glomerulonephritis, but without active lesions. In two other patients (*patients 3* and *5*), no pre-existing diseases or obvious coexisting pathologies were detected.

Most patients (n = 6) were treated with TPE with FFP. Five were treated with C5 complement inhibitor and steroids, one received a steroid and one was treated with intravenous immunoglobulin (IVIg). Two patients had partial or complete recovery of kidney function after treatment, while five patients experienced kidney failure. No deaths were reported.

Functional complement analysis was performed in all seven patients, and four of them showed laboratory evidence of increased activation of the complement system (two had elevated concentrations of C5b9 in the serum, while two had elevated Bb fragment of complement factor B in the serum). Genetic testing for complement-associated abnormalities was performed in four patients: in two cases, no pathogenic genetic variants associated with complement-mediated diseases were identified, while the results of *patients 7* and *8* are still pending.

## 3. Discussion

Since the outbreak of the COVID-19 pandemic in 2019, we have treated a total of eight patients with TMA at our center who presented at the time of COVID-19 infection or shortly thereafter, one of whom presented with TTP and seven others with aHUS. In both groups of patients, we found typical features suggestive of systemic TMA (including MAHA with platelet consumption and organ dysfunction), but the patient with COVID-19-associated TTP was characterized by a lower platelet count and mild neurological symptoms without renal dysfunction, which is consistent with the conventional distinction between TTP and aHUS based on the predominant neurological involvement and more severe thrombocytopenia in TTP and more severe renal involvement in aHUS [12].

Like some other reports [10,13,14], we observed a contrast between clinically mild or even asymptomatic SARS-CoV-2 infection and a severe form of TMA. Namely, all patients with aHUS had severe renal insufficiency with histological signs of extensive vascular and glomerular TMA, moderately to severely swollen endothelia, thickened arterial walls and sometimes completely occluded arterial lumina (Figure 1).

Although there are some reports of patients with TMA after SARS-CoV-2 viremia [15,16,17,18], most patients develop TMA at the time of viremia, which is routinely confirmed by a positive nasopharyngeal PCR smear. Since the beginning of the COVID-19 pandemic and the emergence of more and more patients with oligosymptomatic or even non-respiratory (i.e., gastrointestinal) forms of COVID-19, we have included serologic testing for SARS-CoV-2 in our routine diagnostic protocol to detect a possible recent COVID-19 infection. In our cohort of patients with COVID-19-associated TMA, the patients presented with TTP at the time of viremia, while the patients in the aHUS group presented with TMA that occurred after clearance of SARS-CoV-2, but with serologic evidence of recent COVID-19 infection. Although it is difficult to speculate on the exact time frame between SARS-CoV-2 viremia and the onset of TMA in these patients, the frequency of histopathological findings in the renal biopsies and the severity of the clinical presentation of TMA (severe renal insufficiency, profound hematologic abnormalities) suggest a possible subacute time frame in which the viral load would drop to undetectable levels and seroconversion would occur.

At this point, it is important to emphasize that in most patients with infection-related TMA, infections are probably only one of the many triggers of TMA [19]. Several other non-infectious triggers for TMA, such as certain medications, alloreactivity, malignancies, autoimmune diseases, pregnancy and others, are also known. Two of the patients with aHUS had non-COVID-19 triggers for TMA; namely, one patient was a KTR on immunosuppressive therapy with tacrolimus and everolimus, both of which can be associated with TMA. In another, we detected positive antiphospholipid (aPL) antibodies. In the patient treated with tacrolimus and everolimus, the association with the occurrence of aHUS was less likely as the trough levels of both drugs were within the normal range and neither drug had been introduced recently. However, in the patient with positive aPL antibodies occurring after SARS-CoV-2 infection, we confirmed persistent aPL antibody positivity (on two separate occasions more than 12 weeks apart). Consistent with the clinical manifestation (TMA), he met the classification criteria for the diagnosis of aPS. It is known that there is a high prevalence of aPL antibodies in COVID-19 patients [20], which may only occur transiently during acute COVID-19 infection, but there is still no consensus on their pathogenicity [21,22]. However, they can apparently persist and be an expression of “true” aPS with characteristic clinical manifestations. In fact, patients with COVID-19 have been shown to have a higher risk of developing aPS [23].

Extensive complement activation is another remarkable feature of COVID-19 [24]. It has been suggested that COVID-19 could act as an infectious trigger for complement-mediated aHUS occurring in genetically predisposed patients [13]. Indeed, the first series of renal TMA in COVID-19 patients, which included comprehensive complement work-up, showed that all five patients who underwent genetic testing carried a constitutional complement dysregulation [13]. We observed elevated serum levels of C5b-9 (a marker of terminal complement pathway activation) in two patients in the aHUS subgroup, and elevated serum levels of Bb fragment of complement factor B (a marker of alternative complement pathway activation) in two patients in the aHUS subgroup, which is consistent with the proposed complement activation in COVID-19. Furthermore, the two patients who had elevated serum levels of Bb fragment of complement factor B also showed histological signs of a previously unknown IgA glomerulonephritis without active lesions, but with severe glomerulosclerosis. It has been suggested that complement activity contributes to the pathogenesis of IgA glomerulonephritis, most likely by influencing disease severity [25]. It could be that in these two cases, complement activation played a role in both processes: TMA and the advancement of IgA glomerulonephritis. Patients in our cohort who underwent genetic testing had no pathogenic or probable pathogenic genetic complement abnormalities. However, the results of genetic testing for the two patients with concomitant IgA glomerulonephritis are still pending.

Due to the different underlying pathophysiologies, the treatment of COVID-19-associated TMA requires a specific approach based on the subtype of the syndrome (TTP or aHUS) and possible concomitant triggers (infections, drugs, malignancies, autoimmune diseases, …). However, as TMA is a potentially life-threatening disease, diagnosis should not delay treatment, which should be immediate.

In TTP, first-line therapy is based on daily TPE with FFP, which provides deficient ADAMTS13, with or without steroids [26]. Possible standard adjuvant therapies for TTP are rituximab; the monoclonal antibody against CD20 on the surface of B cells, which inhibits the production of disease-related ADAMTS13 inhibitors by depleting B lymphocytes; and caplacizumab, a humanized bivalent anti-vWF immunoglobulin fragment that inhibits the interaction between vWF multimers and platelets. In addition to urgent TPE, the patient with TTP was also treated with caplacizumab. The latter was chosen instead of rituximab in the context of an active SARS-CoV-2 infection, as caplacizumab has no immunosuppressive properties. The patient with TTP could be discharged after 5 days of therapy with TPE, a steroid and caplacizumab with an improvement in blood count (with a normal platelet count) and no symptoms of the disease.

First-line therapy in aHUS may not be as straightforward as in TTP, although it usually starts with the same initial therapy, i.e., TPE with FFP with or without a steroid. In our previous literature review on COVID-19-associated TMA [10], we showed that slightly more than 50% of patients with COVID-19-associated aHUS received TPE with or without a steroid as an initial therapy, resulting in improvement in about 20%. More than half of the patients received a C5 complement inhibitor (as a monotherapy or after TPE with a steroid failed to improve their condition), with renal function improving in more than 60% of cases.

Six patients with COVID-19-associated aHUS from our cohort were initially treated with TPE with FFP, and five of them received an additional steroid. In 1/6 patients (17%), we observed rapid and complete renal and hematologic recovery after TPE. In a total of 5/6 (83%) patients in our center, their hematologic parameters improved; however, renal function did not recover after the initial treatment with TPE, and we opted for rescue treatment with a C5 complement inhibitor. Unfortunately, all of them developed ESRD despite treatment. In one patient, who was later diagnosed with aPS, the reasonable explanation for the failure of C5 complement inhibitor treatment is the absence of the complement-mediated pathogenesis of TMA. It should be further noted that all patients treated with the C5 complement inhibitor showed pathohistologic signs of extensive vascular and glomerular TMA (two of them with additional features of concomitant collapsing FSGS), and three of them with pathohistologic features of chronic active TMA lesions. Therefore, at least in those three cases (*patients 5*, *7* and *8*) it seems likely that the treatment was unsuccessful because it was given too late in the course of the disease. Interestingly, the only patient with complete recovery of renal function even without C5 complement inhibitor therapy (*patient 3*) was also the only one who presented with a very short history: he had symptoms of COVID-19 ten days prior to presentation with TMA and admission to our center. To summarize, to salvage renal function, it seems that the timing of the treatment is crucial.

Nevertheless, patients with recognized concomitant triggers of TMA (besides SARS-CoV-2 infection), namely positive aPL antibodies and the use of a calcineurin inhibitor (CNI) and everolimus, were later treated according to the disease recommendations, namely anticoagulants and chloroquine for aPS and the optimization of immunosuppressive therapy with the exclusion of everolimus for KTR. The patient with aPS experienced end-stage renal failure, while partial recovery of renal function was observed in KTR.

All patients with COVID-19-associated TMA from this group survived. Although mortality in aHUS has decreased to almost 3% with timely and appropriate treatment [27], the overall mortality in TTP is still high, namely 10–20% [28]. Nevertheless, we strongly believe that in patients with COVID-19-associated TMA, early identification of the TMA subtype, urgent TPE in patients with TTP, and prompt and, if possible, specific treatment in aHUS are crucial to achieve outcomes comparable to the survival statistics for TMA of all causes.

## 4. Patients and Methods

We collected data from patients treated at our center from December 2019 to August 2024. TMA was defined by at least three of the following criteria:Thrombocytopenia (platelet count <150 g/L);MAHA (hemoglobin <100 g/dL, serum lactate dehydrogenase level above upper limit of normal range, undetectable haptoglobin, presence of schistocytes in blood smear);Organ damage (AKI, neurological abnormalities);Features of TMA in renal biopsy.

The association of TMA with COVID-19 was demonstrated either by a proven active infection with SARS-CoV-2 (based on a positive PCR test in an oropharyngeal swab sample) or by a positive serology test for SARS-CoV-2. Serology testing was performed through an enzyme-linked immunosorbent assay for SARS-CoV-2 (Euroimmun AG, Lubeck, Germany) in five patients (*patients 2*, *3*, *4*, *6* and *7*). This test reveals the presence of IgG and IgA antibodies against the S1 subunit of the SARS-CoV-2 spike protein, with optical density values above 1.1 interpreted as positive. During the COVID-19 pandemic, our microbiology laboratory added an additional serology test for SARS-CoV-2; thus, in three patients (*patients 5*, *7* and *8*), serology testing was performed via an electrochemiluminescence immunoassay for SARS-CoV-2 (Elecsys^®^, Roche, Basel, Switzerland). This test reveals the presence of total antibodies (including IgG) against the S1 subunit of the SARS-CoV-2 spike protein (with values above 0.8 BAU/mL interpreted as positive) and against the nucleocapsid protein of SARS-CoV-2 (with optical density values above 1.2 interpreted as positive).

All patients underwent detailed functional complement analysis, including an analysis of the activation of all three pathways (classical, lectin and alternative), the determination of the concentrations of complement components C3 and C4 and complement factors B, H and I, the presence of C3Nef and anti-factor H antibodies, and C5b9 concentrations in the urine and serum.

In addition, genetic testing for variants and complex rearrangements in the genes for the complement (co)factors FH, CFHR1, CFHR2, CFHR3, CFHR4, CHFR5, CD46, FI, FB, C3, DGKE, THBD, PLG, VTN, MASP2 and CD36 was carried out in four patients (*patients 2*, *5*, *7* and *8*) in the aHUS subgroup. However, the results of genetic testing for *patients 7* and *8* are still pending.

The coagulation parameters were normal in all patients, and autoimmune hemolytic anemia was ruled out by a negative Coombs test in all patients. STEC-HUS was ruled out by the absence of Shiga toxin in the stool of all patients.

We actively searched for other known causes or triggers of TMA, such as the following:Certain infections (we performed serologic tests for hepatitis B and C, human immunodeficiency virus, leptospirosis and hantavirus);Autoimmune diseases (we tested for antibodies against cardiolipin, beta2-glycoprotein I and prothrombin, as well as antineutrophil cytoplasmic antibodies and anti-nuclear and anti-glomerular basement membrane antibodies);Malignant diseases;The transplantation of solid organs;Exposure to certain medications that indicate drug-induced TMA (CNI, everolimus, sirolimus, gemcitabine, hydroxychloroquine, …).

## 5. Conclusions

Thrombotic microangiopathy is a significant complication that is not always associated with severe, but often with oligosymptomatic, COVID-19 infection, but has important implications for treatment and patient outcomes.

The pathophysiology of TMA associated with COVID-19 is likely multifactorial. SARS-CoV-2 can either directly damage endothelial cells, leading to inflammation and injury. In patients with pre-existing endothelial injury, local and systemic complement activation and the cytokine storm associated with COVID-19 likely contribute to microangiopathy, with further exacerbation of endothelial injury and thrombosis. SARS-CoV-2 has been shown to trigger complement-mediated aHUS in genetically predisposed patients, induce autoantibody production (e.g., anti-ADAMTS 13 in patients with TTP) or even trigger certain autoimmune diseases (e.g., IgA nephropathy, aPS and others)

Understanding the pathophysiology, clinical manifestations and diagnostic criteria for TMA associated with COVID-19 is crucial for timely intervention, the salvage of renal function and overall patient survival.

## 6. Limitations

The conclusions of the presented case series are based on a retrospective observation of patient data, treatment and clinical course. Patient data were occasionally incomplete, as complement activity was not followed up and genetic complement analysis was not performed in all patients. Further studies are needed to establish a clear cause-and-effect relationship between SARS-CoV-2 infection and the occurrence of TMA.

## Figures and Tables

**Figure 1 ijms-25-12475-f001:**
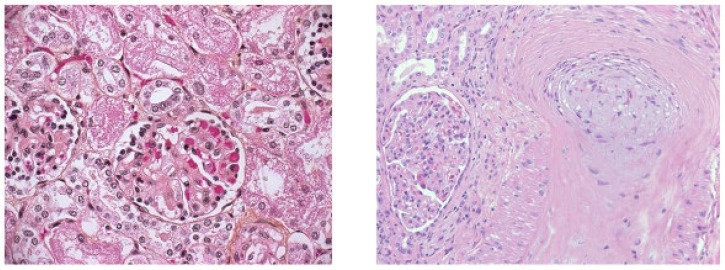
Vascular and glomerular thrombotic microangiopathy in patient with COVID-19-associated TMA, active: in small arteries, endothelium is swollen, lumen is closed, and arterial wall is thickened. Glomerular segmental acute thrombotic microangiopathy with mesangiolysis. Diffuse interstitial edema and bleeding into the interstitium, signs of diffuse tubular damage (hematoxylin and eosin, trichrome staining; 400× magnification). Images provided by Maja Frelih, Institute of Pathology, Medical Faculty, Ljubljana, Slovenia.

**Table 1 ijms-25-12475-t001:** Summary of demographic and clinical characteristics of patients presenting with COVID-19-associated TMA.

Patient Number	Gender	Age	Type of TMA	Comorbidities	Symptoms of COVID-19	PCR for SARS-CoV-2	Serology for SARS-CoV-2	Functional Complement Analysis/Genetic Testing	COVID-19 Vaccination	Lowest Platelet Count (×10^9^/L)	Highest Serum Creatinine (μmol/L)	Kidney Biopsy	Immunoserology	TMA Treatment	Outcome
1	F	35	TTP	none	none	pos.	not performed	normal/-	no	37	77	not performed	neg.	TPE, caplacizumab, steroid	complete hematologic recovery
2	M	43	aHUS	none	none	neg.	pos. anti-S1 (IgG OD 1.14, IgA OD 1.26)	normal/no pathogenic variants	no	67	852 (HD)	glomerular and vascular TMA	pos. lupus anticoagulant and anticardiolipin antibodies (IgM)	TPE, eculizumab, steroid	renal failure
3	M [11]	32	aHUS	none	fatigue, low-grade fever, nausea	neg. (not performed when symptomatic)	pos. anti-S1 (IgG OD 5.54, IgA OD 2.57)	normal/-	no	54	441	not performed	neg.	TPE, steroid	complete recovery of renal function
4	M	33	aHUS	untreated AH, congenital pulmonary valve stenosis	none	neg.	pos. anti-S1 (IgG OD 8.04, IgA OD 5.80)	normal/-	yes (10 months prior to TMA)	29	796 (HD)	glomerular and vascular TMA	neg.	TPE, eculizumab, steroid	renal failure
5	M	25	aHUS	none	none	neg.	pos. anti-S1 (total antibodies 41.96 BAU/mL), pos. anti-N (total antibodies OD > 1.2)	elevated serum C5b-9/no pathogenic variants	no	58	1799 (HD)	chronic TMA, collapsing FSGS	neg.	TPE, eculizumab, steroid	renal failure
6	M	52	aHUS	ESRD due to IgA GN–KTR, DM type 2, AH	none	neg.	pos. anti-S1 (IgG OD 6.29, IgA OD 11.68)	elevated serum C5b-9/-	no	25	2050 (HD)	glomerular and vascular TMA	neg.	TPE, IVIg, discontinuation of everolimus	partial recovery of graft function
7	M	33	aHUS	congenital hearing loss	none	neg.	pos. anti-S1 (total antibodies 1134 BAU/mL; IgG OD 4.00, IgA OD 3.95), pos. anti-N (total antibodies OD > 1.2)	elevated Bb fragment of complement factor B/pending	no	63	1072 (HD)	chronic active TMA, collapsing FSGS, IgA GN	neg.	TPE, eculizumab, steroid	renal failure
8	F	29	aHUS	none	none	neg.	pos. anti-S1 (total antibodies 390.2 BAU/mL), pos. anti-N (total antibodies OD > 1.2)	elevated Bb fragment of complement factor B/pending	no	127	1071 (HD)	chronic active TMA, IgA GN	neg.	ravulizumab, steroid	renal failure

**Legend:** AH, arterial hypertension; aHUS, atypical hemolytic uremic syndrome; anti-N, antibodies against nucleocapsid protein of SARS-CoV-2; anti-S1, antibodies against S1 subunit of the SARS-CoV-2 spike protein; COVID-19, coronavirus disease 2019; DM, diabetes mellitus; ESRD, end-stage renal disease; F, female; FSGS, focal segmental glomerular sclerosis; GN, glomerulonephritis; HD, hemodialysis; IVIg, intravenous immunoglobulins; KTR, kidney transplant recipient; M, male; neg., negative; OD, optical density; PCR, polymerase chain reaction; pos., positive; SARS-CoV-2, severe acute respiratory syndrome coronavirus 2; TMA, thrombotic microangiopathy; TTP, thrombotic thrombocytopenic purpura; TPE, therapeutic plasma exchange.

## Data Availability

The original contributions presented in this study are included in the article. Further inquiries can be directed to the corresponding author.

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
