# Peer review of "Coronavirus Disease 2019-Associated Thrombotic Microangiopathy: A Single-Center Experience"

_ijms, 2024, doi:10.3390/ijms252212475_

Round 1

Reviewer 1 Report

Comments and Suggestions for Authors

The manuscript represents the analysis of seven patients in which thrombotic microangiopathy is observed after a SARS-CoV-2 infection. Even though the manuscript is clinically relevant, the details of the seven patients is scarce. Table 1 is informative; all the patients have TMA, but the conditions of the patients differ significantly, as well as the treatment, giving rise to a lack of consensus on the critical parameters for follow-up, which was unfortunately not performed.  The longitudinal reports of those patients would be essential for the manuscript. The discussion should be enhanced with the reports on thrombocytopenia and complement cascade proteins. It would be interesting to see if the patients had antibodies against PF4. Finally, the limitations, which are several, should be in a separate entity in the manuscript

Author Response

In view of reviewer#1's opinion that the descriptions of disease progression and outcomes in the patients are sparse and that the article would benefit from a longitudinal report, we have now added the exact outcomes and, in part, a date insight into the chronology of the patients' treatment, which we have now clearly presented for all patients in an additional table. We have also added some more data on the dynamics of the hematologic results to the discussion, however, the complement cascade proteins were only determined when the patient was first referred, so the complement dynamics could not be specified. In addition, no patient was tested for PF4 antibodies. As suggested, we have added the limitations in a separate section of the manuscript.

Reviewer 2 Report

Comments and Suggestions for Authors

Very interesting subject, starting from the real observation that there is  a contrast between clinically mild or even asymptomatic SARS-CoV-2 infection and a severe evolution due to the entity called ” thrombotic microangiopathy” (TMA).

The design of the study is very well done, clear, with the detailed deepening of some details related to the infection with the Sarscov2 virus.

Another important observation was related to the fact that all patients with atypical hemolytic–uremic syndrome had severe renal insufficiency with histological signs of extensive vascular and glomerular TMA, moderately to severely swollen endothelia, thickened arterial walls, and sometimes completely occluded arterial lumina. All these modifications could explain the worse evolution in some patients.

It is important to emphasize that in most patients with infection-related TMA, infections are probably only one of the many triggers of the pathological modifications.

Another important thing highlighted by the article is related to the fact that Thrombotic microangiopathy is a significant complication that is not always associated with severe, but often with oligosymptomatic COVID-19 infection, having important  implications for therapeutical plan and patient outcomes.

The pathophysiology of TMA associated with COVID-19 is likely multifactorial.

SARS-CoV-2 can either directly damage endothelial cells, leading to inflammation and  injury. In patients with pre-existing endothelial injury, local and systemic complement  activation and the cytokine storm associated with COVID-19 likely contribute to microangiopathy, with further exacerbation of endothelial injury and thrombosis. SARS-CoV-2  has been shown to trigger complement-mediated atypical hemolytic–uremic syndrome in genetically predisposed patients, induce autoantibody production or  even trigger certain autoimmune diseases.

The article is important in fact that it establishes a connection between Sarscov2 virus infection and thrombotic changes at the level of the microvascular bed, changes that were the basis of multiorgan damage  with a lot of clinical and practical implications.

Understanding the pathophysiology, clinical manifestations and diagnostic criteria  for TMA associated with COVID-19 is crucial for timely intervention, preserve of renal function and overall patient survival.

I have some observations related to the writing of the article such as:

-        In the majority of articles, the section material and method is placed before the results.

-        I suggest to replace the sections for a better understanding: material and methods before the results and disscusions at the end in a logical sequence .

Conclusions are pertinent and logical.

Author Response

In view of reviewer#1's opinion that the descriptions of disease progression and outcomes in the patients are sparse and that the article would benefit from a longitudinal report, we have now added the exact outcomes and, in part, a date insight into the chronology of the patients' treatment, which we have now clearly presented for all patients in an additional table. We have also added some more data on the dynamics of the hematologic results to the discussion, however, the complement cascade proteins were only determined when the patient was first referred, so the complement dynamics could not be specified. In addition, no patient was tested for PF4 antibodies. As suggested, we have added the limitations in a separate section of the manuscript.

As far as it is in line with the journal's rules, we agree that the Methods section should be moved before the Results, as recommended by reviewer#2.

Round 2

Reviewer 1 Report

Comments and Suggestions for Authors

The manuscript has been improved. I have no further comments